# Trend in the Prevalence of Overweight and Obese Adults in São Paulo, Brazil: Analysis between the Years 2006 and 2019

**DOI:** 10.3390/ijerph21040502

**Published:** 2024-04-19

**Authors:** Alisson Padilha de Lima, Ana Paula de Oliveira Barbosa Nunes, Carolina Ferreira Nicoletti, Fabiana Braga Benatti

**Affiliations:** 1School of Medicine, University of Sao Paulo-FMUSP, Sao Paulo 05508-220, SP, Brazil; ana.obnunes@hc.fm.usp.br (A.P.d.O.B.N.); carolina.nicoletti.ferreira@usp.br (C.F.N.); fbenatti@unicamp.br (F.B.B.); 2School of Physical Education, Faculty IELUSC, Joinville 89201-270, SC, Brazil; 3Faculdade de Ciências Aplicadas, Universidade Estadual de Campinas, Limeira 13484-350, SP, Brazil

**Keywords:** epidemiology, health public, overweight, obesity and prevention of adiposity

## Abstract

The aim of this study was to investigate the trend in the prevalence of overweight and obese adults in São Paulo, Brazil, between 2006 and 2019 across chronic diseases and the domains of physical activity. A descriptive retrospective study was carried out on the trend in the prevalence of 26.612 overweight and obese adults (10.150 men and 16.462 women). All data analyzed were based on information from the national system for monitoring risk factors called Protective and Risk Factors for Chronic Diseases by Telephone Survey—VIGITEL. The variables obese and overweight were analyzed in general and stratified by sex, age group, education level, each type of physical activity domain (yes or no), presence of hypertension and diabetes (yes or no), and smoking (yes or no). The prevalence of obesity significantly increased from 11.1% in 2006 to 19.8% in 2019, regardless of age, sex, physical activity practice, and presence of diabetes or hypertension, except for people aged 55–64 y, working people, and smokers. The total prevalence of overweight adults significantly increased overall (from 30.5% in 2006 to 33.4% in 2019) but it significantly increased only in females, in people aged 18–24 y, those who are non-white, those with an education level of 9–11 y, those who are not working, those who are non-smokers, those who did not have diabetes or hypertension, and those who were not physically active during leisure time but physically active at work and at home. There was a significant increase in the prevalence of overweight adults and especially of obese adults living in the city of São Paulo (Brazil) between 2006 and 2019, the latter being observed in nearly every analyzed sub-category, regardless of age, sex, physical activity practice, and presence of diabetes or hypertension.

## 1. Introduction

According to the World Health Organization, being overweight is defined as excessive fat accumulation whereas obesity is defined as excessive fat accumulation to an extent that it may impair health [1]. They are commonly classified according to the body mass index (BMI), a surrogate marker of fatness, with obesity being defined by a BMI ≥ 30 kg/m^2^, and being overweight by a BMI ≥ 25 kg/m^2^ [1,2].

Obesity and being overweight are among the biggest public health problems worldwide, as they can cause premature disability and death by increasing the risk of cardiometabolic diseases (i.e., type 2 diabetes, hypertension, myocardial infarction, stroke, and dyslipidemia) [3,4,5], osteoarthritis, dementia, depression, some types of cancers (i.e., breast, ovarian, prostate, colon, among others) [3,6], and overall mortality [7,8]. In addition, psychological, social, and economic complications can result from obesity, overloading public health systems [9] and decreasing life expectancy [10].

The estimated costs attributable to the main chronic diseases associated with inadequate nutrition show how great of an economic burden these diseases are to the Brazilian Unified Health System [11]. The total costs of hypertension, diabetes, and obesity reached USD 890 million in 2018. However, when separately analyzing obesity as a risk factor for hypertension and diabetes, the costs attributable to obesity itself represented 41% of the total costs [11]. These public health costs are higher in women (56%) [11], as well as the levels of hospitalization resulting from obesity (86.5%) [12].

Alarming data from the World Health Organization (WHO) showed that worldwide obesity has nearly tripled since 1975 [2]. According to the World Obesity Atlas, in 2020, the prevalence of obesity was 14%, and 38% of adults were overweight worldwide [13]. In Brazil, according to the Surveillance of Risk and Protective Factors for Chronic Diseases by Telephone Survey (VIGITEL), there has been a substantial increase in the prevalence of overweight adults, from 42.6% in 2006 to 55.4% in 2019 (a 30% increase), and in the prevalence of obesity, from 11.8% in 2006 to 20.3% in 2019 (a 72% increase) [14]. Importantly, this has paralleled an increase in the prevalence of chronic diseases and a decrease in the practice of physical activity [14].

A similar trend in the prevalence of overweight and obese adults has been observed in São Paulo [14], the most populous city in Brazil, with nearly 11.4 million inhabitants (5.6% of the country’s total population) and a key economic and financial hub in Brazil and Latin America [15]. However, although VIGITEL [14] provides data on the prevalence of chronic non-communicable diseases and physical activity levels, no previous study has reported the temporal trends of overweight and obese adults according to these factors in Brazilian adults living in São Paulo. These data could help to inform health promotion programs and strategies aimed at preventing and treating the increase in the prevalence of overweight and obese adults [16].

Therefore, the aim of this study was to investigate the trend of the prevalence of overweight and obese adults in the city of São Paulo, Brazil, between 2006 and 2019 across chronic diseases and the domains of physical activity.

## 2. Materials and Methods

### 2.1. Study Participants

A descriptive retrospective study was carried out on the trend of the prevalence of overweight and obese adults in the city of São Paulo, Brazil, between the years 2006 and 2019. We chose to analyze data until 2019 due to the onset of the COVID-19 pandemic in 2020, which would act as a very important confounder, being off the scope of this research paper. All data analyzed in this study are based on information from a national system for monitoring risk factors called Protective and Risk Factors for Chronic Diseases by Telephone Survey—VIGITEL. VIGITEL is a population-based study that annually evaluates the adult population (≥18 y) residing in the capitals of the 26 Brazilian states and the Federal District via a telephone surveillance system that conducts a minimum of 2000 adult interviews in each of these cities [17].

The sampling used by the VIGITEL system aims to obtain probabilistic samples of the population of adults residing in households served by at least one fixed telephone line in that year [17]. The VIGITEL system establishes a minimum sample size of 2000 individuals aged 18 or over in each city so that the frequency of any risk factor in the adult population is identified [18]. The description of sampling and application of VIGITEL is thoroughly described by Moura et al. [19].

### 2.2. Research Instruments

The VIGITEL data analyzed in the present study include sex (male; female), age group (in years: 18–24; 25–34; 35–44; 45–54; 55–64; 65 or older), education level (in years of study: 0–8; 9–11; 12 or more); marital status (with or without a partner), race (white or non-white), employment (yes or no), body mass (kg), and height (m). Body mass and height were used to calculate the BMI, which was classified according to the World Health Organization in two categories: excess of body mass (BMI ≥ 25 kg/m^2^), or obesity (BMI ≥ 30 kg/m^2^) [2]. Health-related variables included chronic diseases diabetes and hypertension (yes or no); smoking habits (yes or no); and types of physical activity domains (leisure, transport, work, and domestic).

The variables obesity and being overweight were analyzed in general and stratified by sex, age group, education level, each type of physical activity domain (yes or no), presence of hypertension and diabetes (yes or no), and smoking (yes or no). Leisure time physical activity was analyzed dichotomously (yes or no) according to the question “In the last three months, have you practiced enough physical activity during leisure time? (yes or no); as a form of transportation to and from work or school/university (yes or no); at work (yes or no); and in domestic activities (heavy cleaning/cleaning done alone at home) (yes or no).

The trend in the prevalence of being overweight and obese was estimated for the whole population of São Paulo city during the years 2006 to 2019 in strata of this population defined for each year. In this period, a total of 26.612 individuals were interviewed (Table 1).

### 2.3. Date Analysis

The prevalence of overweight or obese adults was considered a dependent variable and each year of the study was used as an explanatory variable. VIGITEL weighting factors were used to correct sample selection bias. These factors also made it possible to match the distribution of the population studied by VIGITEL—according to sex, age, and schooling—to that identified for the adult population of the city of São Paulo from inter-census projections using a post-stratification procedure.

For the analysis, the time series *Y_t_* is considered, where the times *t* belong to the set {*t*_1_, *t*_2_,…, *t_n_*}. The best-fit line for estimating the temporal trend is defined by the linear regression equation given by *Y_t_* = *b*_0_ + *b*_1_*t* + *e_t_*. In this expression, the parameter *b*_0_ corresponds to a constant, *b*_1_ to the slope of the line, and *e_t_* to a random error [33].

To measure the rate of change of the line that fits the points of the time series, a logarithmic transformation base 10 of the values of *Y_t_* was performed, which contributes to the reduction of the heterogeneity of the variance of the residuals in the linear regression analysis. Moreover, this transformation helps in determining the trend. In the Prais–Winsten method, used for linear regression analysis, the random errors *e_t_* include a first-order temporal autocorrelation structure. In this case, it is assumed that the random errors are given by *e_t_* = *ρe_t−_*_1_ + *w_t_*, where *w_t_* is white noise and *|ρ| <* 1 [33]. Through linear regression, it was possible to estimate the value of the coefficient *b*_1_, applying the confidence interval of this coefficient also for the calculation of the trend or percentage change and the confidence interval of the measure, respectively. The quantitative estimation of the trend was calculated by the following expression: APC = [−1 + 10b^1^] × 100%; and by CI_95%_ = [−1 + 10b^1min^] × 100%; [−1 + 10b^1max^] × 100%. APC refers to the term *annual percent change* and CI to the confidence interval. Significant values (*p* < 0.050) of the regression coefficient indicated an increase or decrease in prevalence [33].

Data processing was carried out using Excel 2010 software (Microsoft Corp., United States), and all analyses were weighted using the rake weight variable and were performed using the “svy” command of Stata software version 12.1 (Stata Corp LP, College Station, TX, USA).

## 3. Results

The results indicate that the prevalence of overweight adults increased by 2.9 percentage points (from 30.5% in 2006 to 33.4% in 2019), a relative increase of 9.6%, while obesity increased by 8.7 percentage points (from 11.1% in 2006 to 19.8% in 2019), a relative increase of 78.4% (Figure 1).

Notably, the prevalence of obesity increased in nearly every analyzed sub-category, regardless of age, sex, physical activity practice, and presence of diabetes or hypertension. The exceptions were people aged 55–64 y, working people, and smokers (Table 2, Table 3 and Table 4).

Although the total prevalence of overweight adults was significantly increased overall, stratifications showed that it was significantly increased only in females, in people aged 18–24 y, non-white people, those with an education level of 9–11 y, and those not working (Table 2).

Regarding chronic diseases and smoking habits, the prevalence of being overweight was significantly increased in non-smokers and in people who did not have diabetes or hypertension (Table 3).

Finally, the prevalence of being overweight was significantly increased in people not physically active during leisure time but physically active at work and at home (Table 4).

## 4. Discussion

Our results, calculated from annual VIGITEL data, identified systematic increases in the overall prevalence of overweight (9.6% relative increase) and especially obese (78.4% relative increase) adults in the city of São Paulo, Brazil, between the years 2006 and 2019. Notably, the increase in the frequency of obese, but not overweight, adults was observed in nearly every analyzed sub-category, regardless of age, sex, physical activity practice, and presence of diabetes or hypertension.

The overall relative increase in the prevalence of obesity was substantially greater than that of being overweight and resembles the sharp increase observed between the early 90s and mid-2000s in countries such as the UK (14.9% in 1993 to 25% in 2006), for instance, only a decade later [34]. Although the occurrence of obesity in 19.8% of adults living in São Paulo in 2019 was still lower than that of developed countries such as the UK (25%) [34], it was higher than that of 12% in 195 countries pooled together [35]. Our data also corroborate a previous study that showed an increase in the frequency of obese and overweight adults from 2006 to 2013 in all regions of Brazil, but a particularly higher increase in the rate of obesity and severe obesity in the southeast region, where Sao Paulo is located [36]. Although São Paulo held the biggest gross domestic product (GDP) in Brazil in 2019 (10.3% of the Brazilian GDP) [37], the conditions of health, life, and prophylaxis for obesity are still precarious in terms of quality public health policies. In this context, these socioeconomic problems may have a direct impact on education levels, food consumption, and adequate levels of physical activity, thus favoring the rise in obesity levels [38,39].

The fact that the overall prevalence of obesity nearly doubled in São Paulo is of great concern, especially considering that this increase reached females and males, from young to old adults, regardless of physical activity practice and the presence of diabetes or hypertension. The one exception regarding age was the 55–64 y age group, where the increase in the occurrence of obesity did not reach statistical significance. Indeed, the greater increases in the rate of obesity were in the younger age groups, from 18 to 44 y, where the relative increases ranged from 117 to 168%, being the greatest in the 18 to 24 y group and decreasing as age increased. This is in line with studies showing that the prevalence of obesity nearly doubles from childhood to young adulthood, but only minor increases are observed in adulthood and older adulthood [40]. The reasons underlying these data are not fully clear but may be associated with a reduced time for physical activity practice and high consumption of “ready-to-eat” ultra-processed foods which normally affect young adults entering higher education and/or the job market [41]. This is of particular concern as obesity is directly related to several metabolic and cardiovascular diseases, which could impose a very high health and economic burden by increasing morbidity and mortality and even potentially affecting life expectancy [7,35,42].

It is interesting that the relative increase in obesity was two times greater in people not working (105%) than in people working (46%), the latter not being statistically significant. The fact that the prevalence of obesity and being overweight increased more sharply in people with higher educational levels may infer that working people with lower educational levels were somehow more protected against excess body mass gain in São Paulo. One may speculate that people with high educational levels in a city such as São Paulo, which serves as a key economic and financial hub not only for the country but also for Latin America, may mostly work in sedentary conditions, in contrast to people with lower educational levels, which could help to explain these results. Indeed, socioeconomic conditions such as education level, race/color, and occupation may influence the occurrence and treatment of obesity [43]. Although greater trends are shown in adults with lower levels of education, increases of greater magnitude have also been observed in those with a higher level of education [44].

The incidence of overweight and obese adults, as well as the trends of increase in this prevalence, were higher in females than in males. This is in line with a previous study that observed a higher increase in the obesity rate in Brazilian women from 2006 to 2019 [45] and corroborates a previous study in the USA showing that obesity frequency increased more in women than in men from 2007–2008 to 2015–2016, thus suggesting a sex-specific vulnerability to weight gain [40]. Some factors may be related to the higher rates of obesity and being overweight among women, such as financial independence, occupations with lower energy expenditure, and reduced time for health care that includes the practice of physical activity and healthy eating habits [46]. This is in line with the work of Florindo et al. [47], who showed a greater trend of low physical activity at work, and during transportation and leisure in Brazilian women when compared to men. Thus, public policies and strategies particularly aimed at promoting a healthy lifestyle in women are of utmost importance to counteract this trend not only in Sao Paulo and Brazil but worldwide.

As expected, the frequency of overweight and obese adults was higher in those with chronic diseases such as diabetes and hypertension throughout the years. These results confirm the strong relationship between increased adiposity and the risk of developing diabetes mellitus, which has been named “Diabesity” [48], and hypertension [49]. Notably, the rise in the prevalence of obesity in adults with chronic diseases may impose even higher levels of morbidity and mortality and overall costs to the health system, highlighting the importance of targeted interventions aimed at these populations [7]. Moreover, although the prevalence of obesity increased in people with and without diabetes and hypertension, the prevalence of overweight only increased significantly in people without these chronic diseases, which may be concerning as this may lead to even further increases in the prevalence of chronic diseases in the long term.

It is of high concern that obesity increased substantially regardless of physical activity practice. Notably, the increase in the prevalence of being overweight was only significant in people not physically active during leisure. Considering that and the fact that the prevalence of obesity is higher in people not active during leisure throughout the years, one may infer that physical activity during leisure may to a certain extent, although not fully, protect against excess body mass gain. These results agree with the systematic review of longitudinal studies by Cleven et al. [50], who reinforce the importance of promoting physical activity in adults and emphasize that only obtaining high levels of physical activity (>300 min/week) can reduce the risk of obesity, cardiovascular diseases, and diabetes [50]. Unexpectedly, however, the prevalence of being overweight increased only in people reporting work-related and domestic physical activity. These results are hard to explain, but they may have occurred due to reporting or memory biases at the time of the interviews, which may overshadow possible associations in the domains of work-related and domestic activities [51]. Another possibility is the low intensity in which physical activity is conducted in work-related and domestic domains, which may have a lower impact on energy balance depending on the amount of physical activity performed [47].

This study is not without limitations. The fact that the interviews were performed by telephone and the data collected were self-reported can generate potential information bias as respondents may have difficulty in accurately recalling information. Notably, the use of fixed telephone lines only may also limit the capability of selecting a representative sample of the population. Finally, because VIGITEL is composed of “yes” or “no” questions, it may not capture the complexity of the respondent’s behavior, particularly regarding physical activity levels.

## 5. Conclusions

In conclusion, there was a significant increase in the prevalence of overweight and especially obese adults living in the city of São Paulo (Brazil) between 2006 and 2019. Notably, the increase in the prevalence of obesity was observed in nearly every analyzed sub-category, regardless of age, sex, physical activity practice, and presence of diabetes or hypertension. This may have a strong impact on public health costs, highlighting the need to implement public health policies aiming at significantly changing lifestyles and promoting physical activity practice and healthy eating habits. Future studies should focus on implementing and evaluating the effectiveness of these policies in the population of Sao Paulo and Brazil.

## Figures and Tables

**Figure 1 ijerph-21-00502-f001:**
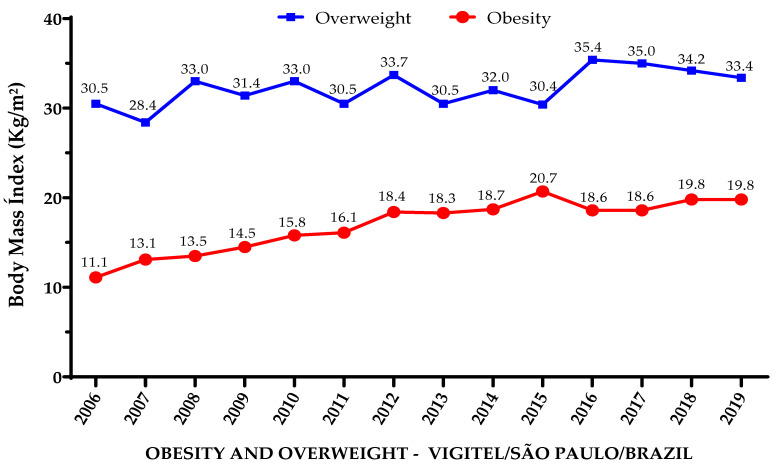
Prevalence of obese and overweight adults living in São Paulo, Brazil, from 2006 to 2019.

**Table 1 ijerph-21-00502-t001:** The total number of lines drawn and eligibility and the number of interviews by sex and total completed, per year, São Paulo, Brazil (VIGITEL: 2006–2019).

Year VIGITEL	Telephone Lines		Number of Interviews Conducted		
	Drawn	Eligible	Men	Women	Total
2006	4.200	3.073	779	1.233	2.012
2007	3.600	2.833	812	1.194	2.006
2008	NS	NS	422	722	1.144
2009	4.800	2.746	757	1.253	2.010
2010	4.200	2.604	748	1.260	2.008
2011	4.000	3.195	780	1.221	2.001
2012	4.400	2.794	679	1.058	1.737
2013	4.400	2.842	775	1.224	1.999
2014	3.200	2.389	588	947	1.535
2015	4.600	2.776	805	1.197	2.002
2016	4.800	2.856	783	1.251	2.034
2017	4.200	2.778	748	1.272	2.020
2018	5.000	2.826	766	1.286	2.052
2019	6.200	2.854	708	1.344	2.052
**Total**	**57.600**	**36.566**	**10.150**	**16.462**	**26.612**

VIGITEL: Surveillance of Risk and Protective Factors for Chronic Diseases by Telephone Survey; NS: not shown. Fonte: VIGITEL BRAZIL [17,20,21,22,23,24,25,26,27,28,29,30,31,32].

**Table 2 ijerph-21-00502-t002:** Annual evolution of the prevalence of obese and overweight adults living in the city of São Paulo according to sociodemographic aspects, VIGITEL, 2006–2019.

Variables	6	7	8	9	10	11	12	13	14	15	16	17	18	19	pp/ano *	*p* **	% Average
**OBESITY**																	
Overall	11.1	13.1	13.5	14.5	15.8	16.1	18.4	18.3	18.7	20.7	18.6	18.6	19.8	19.8	**+0.65 ***	**<0.0001**	16.9
Male	10.1	15.3	13.8	13.6	14.5	14.9	18.5	17.7	20.2	19.9	15.1	17.8	17.9	18.7	**+0.51 ***	**0.004**	16.3
Female	11.7	11.2	13.3	15.2	16.7	17.0	18.3	18.8	17.6	21.3	21.0	19.2	21.4	20.8	**+0.75 ***	**<0.0001**	17.4
**Age (years)**																	
18–24	3.8	3.0	4.5	5.4	4.3	3.3	8.1	4.4	10.0	7.1	8.4	10.2	6.8	10.2	**+0.5 ***	**<0.0001**	6.4
25–34	8.4	12.2	13.6	11.1	12.0	11.5	11.4	13.2	13.1	22.1	16.7	18.5	17.9	18.7	**+0.75 ***	**<0.001**	14.3
35–44	10.5	16.1	13.8	12.4	15.5	22.5	19.9	19.7	22.3	25.3	20.7	19.7	21.3	22.8	**+0.84 ***	**0.003**	18.7
45–54	13.7	17.5	13.5	19.3	20.7	19.6	21.3	24.6	20.6	24.8	23.7	19.6	22.1	21.3	**+0.57 ***	**0.008**	20.2
55–64	16.9	14.3	24.6	23.7	19.6	21.6	27.0	27.9	21.5	22.8	18.4	19.2	26.0	21.8	0.28	0.363	21.8
65 ou +	17.4	14.9	12.9	17.8	21.4	18.7	21.4	22.9	20.2	18.6	21.3	19.2	24.3	22.1	**+0.51 ***	**0.009**	19.5
**Educational Level**																	
0–8 years	15.9	16.2	15.1	16.3	20.1	18.8	21.0	23.9	20.9	26.8	22.1	20.2	21.7	21.4	**+0.56 ***	**0.012**	20.0
9–11 years	6.6	11.6	12.0	14.0	13.0	15.0	17.5	13.9	20.2	19.4	16.6	18.9	20.6	20.1	**+0.85 ***	**<0.0001**	15.7
12 or more	8.6	10.7	13.2	12.6	13.7	13.4	16.3	15.4	13.5	15.5	16.9	16.4	17.1	18.0	**+0.58 ***	**<0.0001**	14.4
**Marital Status**																	
No Partner	8.3	10.1	9.9	13.1	12.0	13.4	17.0	12.8	14.1	16.8	17.6	17.1	14.2	17.1	**+0.61 ***	**<0.0001**	13.8
With Partner	13.4	15.6	17.1	15.9	18.8	18.7	19.5	22.8	21.8	24.1	19.5	19.7	24.5	22.2	**+0.66 ***	**<0.0001**	19.5
**Race**																	
White	9.8	13.4	13.5	13.5	14.5	18.2	19.4	18.1	19.5	18.6	19.9	16.3	20.9	16.9	**+0.58 ***	**0.003**	16.6
No White	12.3	12.7	13.4	15.7	17.1	13.1	15.4	17.5	16.8	22.8	16.7	20.2	18.2	23.7	**+0.67 ***	**<0.0001**	16.8
**Work**																	
Yes	14.6	14.0	15.2	18.1	20.6	22.8	23.1	21.3	21.9	23.2	22.3	19.9	22.0	21.3	0.53	0.058	20.0
No	9.3	12.6	12.7	12.5	13.3	13.1	16.2	16.8	16.9	19.3	16.8	17.8	18.7	19.1	**+0.69 ***	**<0.0001**	15.4
**OVERWEIGHT**																	
Overall	30.5	28.4	33.0	31.4	33.0	30.5	33.7	30.5	32.0	30.4	35.4	35.0	34.2	33.4	**0.31**	**0.003**	32.2
Male	35.3	34.7	40.3	38.4	42.2	37.6	37.7	35.9	36.1	36.0	42.3	37.7	37.9	37.3	0.06	0.713	37.8
Female	27.5	23.7	27.2	26.8	27.0	25.0	31.1	26.0	29.3	25.7	30.7	33.0	31.0	30.1	**0.46**	**<0.0001**	28.2
**Age (years)**																	
18–24	16.3	12.3	20.1	15.9	17.1	11.2	12.2	19.2	19.1	17.9	17.7	22.4	23.4	17.7	**0.46**	**0.048**	17.3
25–34	20.7	27.0	31.9	29.6	31.6	30.6	35.5	23.9	32.6	22.8	36.5	31.7	33.3	31.0	0.40	0.083	29.9
35–44	34.2	32.0	33.8	33.9	34.0	32.2	32.7	33.4	30.6	32.1	39.9	37.2	36.5	34.6	0.23	0.224	34.1
45–54	40.5	33.6	36.5	37.7	35.5	35.2	37.9	38.1	34.8	37.2	33.6	39.0	40.4	41.3	0.2	0.220	37.2
55–64	43.4	35.7	34.3	35.1	36.5	36.5	34.5	39.0	35.8	37.7	42.1	34.8	33.7	37.1	−0.08	0.663	36.9
65 ou +	33.1	34.2	43.1	35.6	39.4	36.9	39.0	34.3	34.8	33.6	38.0	36.9	35.0	37.0	0.12	0.261	36.5
**Educational Level**																	
0–8 years	35.8	29.9	32.1	33.8	34.7	34.2	36.0	33.0	32.9	32.5	34.3	34.1	35.4	34.9	0.12	0.261	33.8
9–11 years	25.7	24.4	32.4	27.8	31.2	25.9	28.7	28.4	31.9	28.0	36.4	36.3	32.7	31.6	**0.60**	**0.003**	30.1
12 or more	26.5	32.3	35.3	33.0	32.8	31.4	37.4	29.5	30.6	30.8	35.6	34.4	34.4	33.8	0.23	0.205	32.7
**Marital Status**																	
No Partner	25.5	23.1	28.3	26.9	27.5	23.2	28.0	24.1	29.5	26.0	30.4	30.8	29.8	28.4	**0.39**	**0.0001**	27.3
With Partner	34.8	33.1	37.5	35.9	36.8	37.2	38.2	36.8	33.9	34.7	39.7	38.0	37.9	38.5	0.24	0.057	36.6
**Race**																	
White	31.8	28.6	33.3	35.0	31.6	30.8	35.3	31.8	32.6	31.6	35.5	36.4	32.5	33.7	0.23	0.063	32.9
No White	29.1	28.4	32.6	27.6	34.4	30.5	33.4	29.2	31.8	28.8	34.6	34.7	36.7	32.9	**0.42**	**0.002**	31.8
**Work**																	
Yes	30.3	29.1	34.0	32.7	33.7	30.8	34.3	30.8	31.7	31.2	35.4	37.1	35.5	34.1	**0.34**	**0.023**	32.9
No	30.8	27.0	30.9	29.4	31.7	29.9	32.6	30.2	32.5	28.9	35.5	31.3	31.4	21.9	**0.02**	**0.883**	30.3

* PP/Year: prevalence per year; ** *p*-value obtained by Prais–Winsten regression.

**Table 3 ijerph-21-00502-t003:** Annual evolution of the prevalence of obese and overweight adults living in the city of São Paulo according to chronic diseases and smoking habits. VIGITEL. 2006–2019.

	6	7	8	9	10	11	12	13	14	15	16	17	18	19	pp/ano *	*p* **	% Average
**OBESITY**																	
**Diabetes**																	
Yes	25.2	25.5	26.4	31.1	30.5	30.3	37.2	35.7	34.1	30.0	33.0	30.1	38.5	39.7	**+0.89 ***	**0.005**	32.0
No	10.1	12.2	12.4	13.2	14.2	14.8	15.8	16.7	16.9	19.5	16.8	17.3	18.2	18.0	**+0.60 ***	**<0.0001**	15.4
**Hypertension**																	
Yes	23.3	23.3	23.9	30.3	29.3	28.8	34.4	35.0	30.1	31.4	31.4	31.9	31.8	30.9	**+0.62 ***	**0.034**	29.7
No	7.2	9.9	9.3	8.8	11.2	11.8	12.7	12.7	13.9	16.5	13.7	13.2	15.9	16.3	**+0.63 ***	**<0.0001**	12.4
**Smoking**																	
Yes	9.0	10.6	10.5	13.6	11.6	14.9	14.7	20.8	15.3	11.3	10.7	17.9	12.6	12.4	0.26	0.297	13.3
No	11.5	13.6	14.2	14.7	16.6	16.3	19.1	17.8	19.2	22.2	19.8	18.7	20.7	20.9	**+0.69 ***	**<0.0001**	17.5
**OVERWEIGHT**																	
**Diabetes**																	
Yes	35.6	27.7	34.6	33.6	42.2	43.2	43.0	36.3	28.3	39.3	38.0	34.6	35.5	32.2	−0.23	0.955	36.0
No	30.2	28.5	32.8	31.2	32.0	29.5	32.5	29.9	32.5	29.6	35.1	34.9	34.1	33.5	**0.33**	**0.002**	31.9
**Hypertension**																	
Yes	37.5	35.8	38.4	34.9	36.5	37.1	35.7	37.2	37.1	38.6	40.5	38.4	36.7	32.9	−0.06	0.721	36.9
No	28.2	26.2	30.7	30.1	31.8	28.2	32.8	28.5	29.9	27.2	33.5	33.5	33.3	33.6	**0.38**	**0.007**	30.5
**Smoking**																	
Yes	32.7	25.9	25.3	27.3	34.1	27.8	31.6	27.1	28.0	29.0	37.1	27.0	29.2	36.2	0.30	0.13	29.9
No	30.0	29.0	34.8	32.2	32.8	31.0	34.1	31.2	32.6	30.6	35.2	36.2	34.9	33.0	**0.28**	**0.022**	32.7

* PP/Year: prevalence per year; ** *p*-value obtained by Prais–Winsten regression.

**Table 4 ijerph-21-00502-t004:** Annual evolution of the prevalence of obese and overweight adults living in the city of São Paulo according to the type and number of physical activity domains. VIGITEL. 2006–2019.

	6	7	8	9	10	11	12	13	14	15	16	17	18	19	pp/ano	*p* **	% Average
**OBESITY**																	
**Leisure**																	
Yes	9.9	10.9	11.0	11.2	12.1	13.2	16.1	14.5	18.2	16.8	14.7	13.2	17.3	16.0	**+0.50 ***	**0.003**	13.9
No	11.9	14.7	15.5	16.9	18.1	18.3	20.2	21.2	19.1	24.1	22.1	23.3	22.5	24.0	**+0.84 ***	**<0.0001**	19.4
**Transport**																	
Yes	NS	NS	NS	11.0	11.9	13.2	14.5	14.7	14.5	17.3	15.4	14.6	16.7	19.7	**+0.66 ***	**0.001**	14.9
No	NS	NS	NS	15.0	15.4	13.2	18.3	19.7	20.6	22.5	18.9	22.1	21.7	18.3	**+0.62 ***	**0.032**	18.7
**Work-Related**																	
Yes	9.8	12.4	11.9	11.3	13.7	13.9	17.3	16.0	16.9	17.5	16.3	16.6	17.2	18.8	**+0.61 ***	**<0.0001**	15.0
No	8.7	12.8	14.0	14.5	13.0	12.0	14.4	18.3	17.0	22.2	17.5	20.3	21.3	19.5	**+0.83 ***	**<0.0001**	16.1
**Domestic**																	
Yes	11.9	12.8	12.5	14.2	17.6	17.4	18.1	18.8	19.0	23.6	19.1	18.5	20.3	19.3	+0.65 *	0.002	17.4
No	10.2	13.3	14.5	15.0	13.7	14.7	18.8	17.8	18.4	18.1	18.0	18.6	19.0	20.4	+0.64 *	<0.0001	16.5
**OVERWEIGHT**																	
**Leisure**																	
Yes	33.0	32.7	32.5	35.4	37.1	34.8	37.0	33.3	32.5	32.6	38.2	37.7	35.2	34.5	0.17	0.302	34.8
No	28.9	25.4	33.3	28.8	30.5	27.5	31.4	28.5	31.6	28.6	32.9	32.7	33.1	32.3	**0.35**	**0.001**	30.4
**Transport**																	
Yes	NS	NS	NS	30.6	31.2	27.7	34.8	27.0	29.5	28.9	33.1	35.0	34.4	33.7	0.46	0.062	31.4
No	NS	NS	NS	36.1	36.9	35.2	34.0	35.6	35.3	33.7	38.8	39.8	37.1	34.7	0.10	0.674	36.1
**Work-Related**																	
Yes	28.7	27.1	33.9	34.2	33.4	29.1	32.6	30.3	33.7	32.0	36.0	35.2	34.7	33.8	**0.39**	**0.019**	32.5
No	32.8	32.6	34.4	30.0	33.9	33.6	37.2	31.2	28.4	29.4	34.6	40.6	36.9	34.6	0.24	0.398	33.6
**Domestic**																	
Yes	28.8	25.2	30.3	29.1	29.7	27.3	34.0	28.5	31.4	29.8	33.8	33.6	33.3	32.8	**0.50**	**<0.0001**	30.5
No	32.3	31.7	35.6	34.3	36.8	34.0	33.3	32.4	32.7	30.9	37.5	37.2	35.3	34.3	0.15	0.363	34.2

* PP/Year: prevalence per year; ** *p*-value obtained by Prais–Winsten regression.

## Data Availability

The data analyzed in this research are publicly available in the database of the Brazilian Ministry of Health portal (https://svs.aids.gov.br/bases_vigitel_viva/vigitel.php (accessed on March 2022)). Any additional questions can be sent to the correspondence author.

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
