# Peer review of "Trend in the Prevalence of Overweight and Obese Adults in São Paulo, Brazil: Analysis between the Years 2006 and 2019"

_ijerph, 2024, doi:10.3390/ijerph21040502_

Round 1
Reviewer 1 Report
Comments and Suggestions for Authors
This study was well designed. The writing is fluid and its results are important. I have few considerations. On the other hand, I see the need to explain in some topic of the study why the cut was made in 2019.
Introduction
-The introduction is fluid, with excellent foundation. I would only add information about São Paulo's tax collection and its economic impact on Latin America. This can even be taken into consideration when discussing the increase in cases of overweight and obesity (relationship between work, leisure and health).
-Avoid the repetition of the name VIGITEL in the introduction.
Methods
-It is not clear what the term CAPITAL means.
-Change “gender” by “sex” (here and throughout the manuscript)
-Change “weight” by “body mass” (here and throughout the manuscript)
-In the legend of Table 1 the term VIGITEL appears in Portuguese, but it the text appears in English
Results
-Be consistent with the font type in figures and in the text.
-Be consistent with ISI for units (e.g BMI).
Discussion
- The term “prevalence” appears very closely and subsequently in some paragraphs. Improve this description.
Author Response
This study was well designed. The writing is fluid and its results are important. I have few considerations. On the other hand, I see the need to explain in some topic of the study why the cut was made in 2019.
Thank you for the important comment. The cut was made in 2019, because the onset of COVID-19 in 2020 was a major factor influencing the trends of overweight and obesity in São Paulo and worldwide, which would be off the scope and change the main aim of the study to explore these trends in overall normal conditions throughout the years. This has been added to the manuscript for clarity purposes.
Introduction
-The introduction is fluid, with excellent foundation. I would only add information about São Paulo's tax collection and its economic impact on Latin America. This can even be taken into consideration when discussing the increase in cases of overweight and obesity (relationship between work, leisure and health).
We have added a sentence in the introduction and discussed the data considering this information.
-Avoid the repetition of the name VIGITEL in the introduction.
This has been changed accordingly.
Methods
-It is not clear what the term CAPITAL means.
This has been changed accordingly.
-Change “gender” by “sex” (here and throughout the manuscript)
This has been changed accordingly.
-Change “weight” by “body mass” (here and throughout the manuscript)
This has been changed accordingly.
-In the legend of Table 1 the term VIGITEL appears in Portuguese, but it the text appears in English
This has been corrected.
Results
-Be consistent with the font type in figures and in the text.
This has been corrected.
-Be consistent with ISI for units (e.g BMI).
This has been corrected.
Discussion
- The term “prevalence” appears very closely and subsequently in some paragraphs. Improve this description.
This has been changed accordingly.
Reviewer 2 Report
Comments and Suggestions for Authors
The article entitled “Trend in the prevalence of overweight and obesity in adults in São Paulo-SP, Brazil: Analysis between the years 2006 to 2019” presents a compelling exploration of the prevalence of overweight and obesity in São Paulo. The authors delve into the higher numbers of people living with overweight and obesity in São Paulo, Brazil, shedding light on the impact of public health and the relation with the increase in chronic diseases. While the research contributes significantly to our understanding of the prevalence of obesity and overweight, certain aspects warrant further examination and discussion.
1. Introduction
Please, consider providing a brief definition of overweight in addition to obesity.
Rephrasing the final sentence for clarification. It could be better to explicitly state that the study aims to contribute valuable insights to inform health promotion programs and strategies aimed at addressing the rise in overweight and obesity.
2. Results and Discussion
The authors should discuss the increase in obesity prevalence in São Paulo in comparison with international trends, particularly with countries experiencing similar rises in obesity rates.
It would be insightful if the authors explored the age-specific trends in obesity prevalence, emphasising the significant increases in younger age groups. They should discuss potential factors influencing these greater relative increases in these age brackets.
The authors should emphasise the implications of the association between overweight and obesity with chronic diseases, discussing the potential public health impact of rising prevalence rates in individuals with these conditions and the importance of targeted interventions. It would be interesting, to include data from the public health system in Brazil concerning these conditions in the discussion as well.
Considering the widespread prevalence of mobile phones in today’s society, the reliance on fixed telephone lines as the primary mode of contact may present a potential limitation. Mobile phones have become ubiquitous, and an increasing number of individuals rely solely on them for communication. Given this, could the reliance on fixed telephone lines in the VIGITEL system introduce a bias or limitation in capturing a representative sample of the population, particularly among those who primarily use mobile phones? I would appreciate the authors' insights on how this potential limitation may impact the generalisability of their findings.
Conclude the discussion by providing recommendations for future research. Identify areas where further investigation is needed, and the effectiveness of public health interventions in addressing the obesity epidemic in São Paulo (and possibly across Brazil).
Author Response
The article entitled “Trend in the prevalence of overweight and obesity in adults in São Paulo-SP, Brazil: Analysis between the years 2006 to 2019” presents a compelling exploration of the prevalence of overweight and obesity in São Paulo. The authors delve into the higher numbers of people living with overweight and obesity in São Paulo, Brazil, shedding light on the impact of public health and the relation with the increase in chronic diseases. While the research contributes significantly to our understanding of the prevalence of obesity and overweight, certain aspects warrant further examination and discussion.
1. Introduction
Please, consider providing a brief definition of overweight in addition to obesity.
This has been changed accordingly.
Rephrasing the final sentence for clarification. It could be better to explicitly state that the study aims to contribute valuable insights to inform health promotion programs and strategies aimed at addressing the rise in overweight and obesity.
This has been changed accordingly.
- Results and Discussion
The authors should discuss the increase in obesity prevalence in São Paulo in comparison with international trends, particularly with countries experiencing similar rises in obesity rates.
We have extended the discussion in comparison with countries such as the US and UK.
It would be insightful if the authors explored the age-specific trends in obesity prevalence, emphasising the significant increases in younger age groups. They should discuss potential factors influencing these greater relative increases in these age brackets.
We have further emphasized the more significant increases in younger age groups and compared it with previous studies.
The authors should emphasise the implications of the association between overweight and obesity with chronic diseases, discussing the potential public health impact of rising prevalence rates in individuals with these conditions and the importance of targeted interventions. It would be interesting, to include data from the public health system in Brazil concerning these conditions in the discussion as well.
We have added a sentence addressing this very important comment, thank you.
Considering the widespread prevalence of mobile phones in today’s society, the reliance on fixed telephone lines as the primary mode of contact may present a potential limitation. Mobile phones have become ubiquitous, and an increasing number of individuals rely solely on them for communication. Given this, could the reliance on fixed telephone lines in the VIGITEL system introduce a bias or limitation in capturing a representative sample of the population, particularly among those who primarily use mobile phones? I would appreciate the authors' insights on how this potential limitation may impact the generalisability of their findings.
We agree with the reviewer and have added this as a limitation.
Conclude the discussion by providing recommendations for future research. Identify areas where further investigation is needed, and the effectiveness of public health interventions in addressing the obesity epidemic in São Paulo (and possibly across Brazil).
We have included a sentence in the conclusion. Thank you.
Reviewer 3 Report
Comments and Suggestions for Authors
The study epidemiologically analyzes the issue of overweight and obesity in the city of Sao Paul. The study is potentially interesting, but does not appear to be groundbreaking. In general, many aspects of chronic degenerative diseases have not yet been investigated, much less any sex differences; therefore, since these data are present in the study, and in some cases are significant, more unexplored and innovative findings that can help the field of research should be highlighted in general.
In addition, here are some other suggestions for authors:
Introduction:
"Obesity is defined as excessive accumulation of fat and is commonly classified by a body mass index (BMI) ≥30 kg/m2 [1,2]." what does the authors mean with excessive accumulation of fat? Please specify it.
"This data could help to inform health promotion programs that act on lifestyle changes which can aid the non-pharmacological treatment and prophylaxis against the increase in the prevalence of excess weight and chronic diseases [15]." Do the authors intend an excess of weight or of fat? Please make it clear.
Materials and Methods:
"The VIGITEL data analyzed in the present study includes gender (male; female), age group (in years: 18-24; 25-34; 35-44; 45-54; 55-64; 65 or more), education level (in years of study: 0-8; 9-11; 12 or more); marital status (with or without a partner), race (white or non-white), employment (Yes or No), weight (kg) and height (m)." Are the weight and height values measured by trained professionals or reported from the sample? It is important to highlight that, the data reported from the sample, represents a limitation of the study.
"Weight and height were used to calculate the BMI which was classified according to two categories: excess of weight (BMI ≥ 25kg/m2) or obesity (BMI ≥ 30kg/m2). " BMI classification based on what criteria was made? Please provide a reference for this.
Discussion:
"The prevalence of overweight and obesity as well as the trends of increase in this prevalence were higher in females than in males. Some factors may be related to the higher rates of obesity and overweight among women, such as financial independence, occupations with lower energy expenditure, and reduced time for health care that include the practice of physical activity and healthy eating habits [40]. This is in line with Florindo et al. [41] who showed a greater prevalence of low physical activity at work, and during transportation and leisure in Brazilian women when compared to men." This result is certainly one of the most innovative. More discussion of the topic should be expanded and put more emphasis on it.
Author Response
The study epidemiologically analyzes the issue of overweight and obesity in the city of Sao Paul. The study is potentially interesting, but does not appear to be groundbreaking. In general, many aspects of chronic degenerative diseases have not yet been investigated, much less any sex differences; therefore, since these data are present in the study, and in some cases are significant, more unexplored and innovative findings that can help the field of research should be highlighted in general.
In addition, here are some other suggestions for authors:
Introduction:
"Obesity is defined as excessive accumulation of fat and is commonly classified by a body mass index (BMI) ≥30 kg/m2 [1,2]." what does the authors mean with excessive accumulation of fat? Please specify it.
We have changed the sentence for clarity purposes.
"This data could help to inform health promotion programs that act on lifestyle changes which can aid the non-pharmacological treatment and prophylaxis against the increase in the prevalence of excess weight and chronic diseases [15]." Do the authors intend an excess of weight or of fat? Please make it clear.
We have rephrased the sentence for clarity purposes as this was also highlighted by another reviewer, thank you for the comment.
Materials and Methods:
"The VIGITEL data analyzed in the present study includes gender (male; female), age group (in years: 18-24; 25-34; 35-44; 45-54; 55-64; 65 or more), education level (in years of study: 0-8; 9-11; 12 or more); marital status (with or without a partner), race (white or non-white), employment (Yes or No), weight (kg) and height (m)." Are the weight and height values measured by trained professionals or reported from the sample? It is important to highlight that, the data reported from the sample, represents a limitation of the study.
We have added this as a limitation to the manuscript.
"Weight and height were used to calculate the BMI which was classified according to two categories: excess of weight (BMI ≥ 25kg/m2) or obesity (BMI ≥ 30kg/m2). " BMI classification based on what criteria was made? Please provide a reference for this.
We have added this information.
Discussion:
"The prevalence of overweight and obesity as well as the trends of increase in this prevalence were higher in females than in males. Some factors may be related to the higher rates of obesity and overweight among women, such as financial independence, occupations with lower energy expenditure, and reduced time for health care that include the practice of physical activity and healthy eating habits [40]. This is in line with Florindo et al. [41] who showed a greater prevalence of low physical activity at work, and during transportation and leisure in Brazilian women when compared to men." This result is certainly one of the most innovative. More discussion of the topic should be expanded and put more emphasis on it.
We have expanded this discussion as requested.
Reviewer 4 Report
Comments and Suggestions for Authors
This is a time series study that used data from the Vigitel telephone survey to the temporal trend in the prevalence of overweight and obesity in the adult population of São Paulo from 2006 to 2019.
Although not innovative, the study provides important information about the trend of overweight and obesity in the most populous city in Brazil.
Data from other studies using Vigitel indicate that São Paulo has one of the highest trends in severe obesity when compared to other Brazilian capitals. It would be interesting to include this information in the discussion.
In the discussion, it would be interesting to compare the results of trends in overweight and obesity with other Brazilian regions. There are recent studies that used Vigitel data in other Brazilian capitals.
Some important limitations were not considered. For example: in the period from 2006 to 2019, it is possible that fixed telephone coverage in São Paulo varied. As Vigitel only uses landline telephony, it is important to mention these as limitations of the study.
Despite this, a studies that compared data from the National Health Survey (in person) with data from Vigitel (landline) observed that there was no significant statistical difference in the prevalence of overweight,
Author Response
This is a time series study that used data from the Vigitel telephone survey to the temporal trend in the prevalence of overweight and obesity in the adult population of São Paulo from 2006 to 2019.
Although not innovative, the study provides important information about the trend of overweight and obesity in the most populous city in Brazil.
Data from other studies using Vigitel indicate that São Paulo has one of the highest trends in severe obesity when compared to other Brazilian capitals. It would be interesting to include this information in the discussion. In the discussion, it would be interesting to compare the results of trends in overweight and obesity with other Brazilian regions. There are recent studies that used Vigitel data in other Brazilian capitals.
We have included this information and expanded this discussion as requested.
Some important limitations were not considered. For example: in the period from 2006 to 2019, it is possible that fixed telephone coverage in São Paulo varied. As Vigitel only uses landline telephony, it is important to mention these as limitations of the study.
We have added this limitation to the manuscript, thank you.
Despite this, a studies that compared data from the National Health Survey (in person) with data from Vigitel (landline) observed that there was no significant statistical difference in the prevalence of overweight,
It appears that the sentence was incomplete, which made changes impossible.
Reviewer 5 Report
Comments and Suggestions for Authors
In this manuscript, Alisson Padilha de Lima et al investigated the trend of the prevalence of overweight and obesity in the adult population of the city of São Paulo-SP, Brazil, between 2006 and 2019, across chronic diseases and the domains of physical activity. The authors carried out a retrospective study through analyzing the information from a national system for monitoring risk factors called Protective and Risk Factors for Chronic Diseases by Telephone Survey – VIGITEL.
The authors found the prevalence of obesity significantly increased from 11.1% in 2006 to 19.8% in 2019, regardless of age, sex, physical activity practice, and presence of diabetes or hypertension, except for people aged 55-64, working people, and smokers. More importantly, they reported there was a significant increase in the prevalence of overweight and especially obesity in adults living in the city of São Paulo (Brazil) between 2006 and 2019, the latter being observed in nearly every analyzed sub-category, regardless of age, sex, physical activity practice and presence of diabetes or hypertension.
Overall, this is a well-designed study with sound data analysis, revealing an important trend of overweight and obesity in the adult population of the city of São Paulo-SP, Brazil. Besides, the manuscript is clear and well-written.
A few specific points need to be addressed before publication
1. The method section regarding data analysis requires more detailed descriptions, for example, can authors elaborate on the specific considerations and steps taken to control autocorrelation in the Prais-Winsten regression models?
2. How do the VIGITEL weighting factors contribute to correcting sample selection bias, and what is the rationale behind using them in the analysis?
3. Are there any inherent limitations in the study design or methodology that should be acknowledged? How might these limitations affect the findings?
Author Response
In this manuscript, Alisson Padilha de Lima et al investigated the trend of the prevalence of overweight and obesity in the adult population of the city of São Paulo-SP, Brazil, between 2006 and 2019, across chronic diseases and the domains of physical activity. The authors carried out a retrospective study through analyzing the information from a national system for monitoring risk factors called Protective and Risk Factors for Chronic Diseases by Telephone Survey – VIGITEL.
The authors found the prevalence of obesity significantly increased from 11.1% in 2006 to 19.8% in 2019, regardless of age, sex, physical activity practice, and presence of diabetes or hypertension, except for people aged 55-64, working people, and smokers. More importantly, they reported there was a significant increase in the prevalence of overweight and especially obesity in adults living in the city of São Paulo (Brazil) between 2006 and 2019, the latter being observed in nearly every analyzed sub-category, regardless of age, sex, physical activity practice and presence of diabetes or hypertension.
Overall, this is a well-designed study with sound data analysis, revealing an important trend of overweight and obesity in the adult population of the city of São Paulo-SP, Brazil. Besides, the manuscript is clear and well-written.
A few specific points need to be addressed before publication
1. The method section regarding data analysis requires more detailed descriptions, for example, can authors elaborate on the specific considerations and steps taken to control autocorrelation in the Prais-Winsten regression models?
2. How do the VIGITEL weighting factors contribute to correcting sample selection bias, and what is the rationale behind using them in the analysis?
We modified the sentence to make it clearer and answer the two questions about the statistical methods used.
3. Are there any inherent limitations in the study design or methodology that should be acknowledged? How might these limitations affect the findings?
We have expanded our paragraph on limitations that include the ones regarding the methods and design.
Round 2
Reviewer 2 Report
Comments and Suggestions for Authors
I appreciate the effort and dedication the authors have put into addressing the comments and suggestions.
I am now satisfied with the current state of the manuscript and believe it is well-prepared for the next steps in the publication process.
Author Response
I appreciate the effort and dedication the authors have put into addressing the comments and suggestions.
I am now satisfied with the current state of the manuscript and believe it is well-prepared for the next steps in the publication process.
We appreciate your valuable input in the peer review process. Thank you very much.
Reviewer 3 Report
Comments and Suggestions for Authors
The version revised by the authors, is greatly improved in methodology and discussion.
However, reworking the introduction makes it even less clear how excess fat, and not so much BMI, is related to the whole pathological picture egregiously illustrated.
Therefore, the authors should revise the introduction, stressing and emphasizing the concept of excess fat mass and health.
Author Response
We thank the reviewer for the comment. We have extended the explanation and we hope that it may now more clrearly reflect the relationship between body fat and health.
